# What motivates consumers to buy organic foods? Results of an empirical study in the United States

**Raghava R. Gundala**[1]**, Anupam Singh**[2]*

**1** Department of Business, University of Wisconsin-Parkside, Kenosha, Wisconsin, United States of America,
**2** Faculty of Economic Sciences and Management, Nicolaus Copernicus University, Torun, Poland

* anupam.nitb@gmail.com

## Abstract

Consumers perceive organic foods as more nutritious, natural, and environmentally friendly than non-organic or conventional foods. Since organic foods developed, studies on consumer behavior and organic foods have contributed significantly to its development. The preseent study aims to identify the factors affecting consumer buying behaviour toward organic foods in the United States. Survey data are collected from 770 consumers in the Midwest, United States. ANOVA, multiple linear regression, factor analysis, independent t-tests, and hierarchical multiple regression analysis are used to analyze the collected primary data. This research confirms health consciousness, consumer knowledge, perceived or subjective norms, and perception of price influence consumers' attitudes toward buying organic foods. Availability is another factor that affected the purchase intentions of consumers. Age, education, and income are demographic factors that also impact consumers' buying behavior. The findings help marketers of organic foods design strategies to succeed in the US's fast-growing organic foods market.

**Data Availability Statement:** All relevant data are within the manuscript and its Supporting Information files.

**Funding:** The author(s) received no specific funding for this work.

## Introduction

### What is organic food?

Foods that are cultivated without the application of chemical pesticides can be called organic foods [1]. The feed cannot include antibiotics or growth hormones for the food products labeled organic for foods derived from animals (e.g., eggs, meat, milk, and milk products) [2]. Organic foods are perceived as environmentally safe, as chemical pesticides and fertilizers are not used in their production. They also are not grown from genetically modified organisms. Furthermore, organic foods are not processed using irradiation, industrial solvents, or synthetic food additives [3]. Thus, these foods are considered environmentally safe, as they are produced using ecologically sound methods.

When the world's population was low, almost all agriculture was primarily organic and near-natural. However, these traditional practices, passed from one generation to the next, did not produce enough food to meet the rapidly increasing global population's demands. This led

**Competing interests:** The authors have declared that no competing interests exist.

to the "green revolution," in which farmers used technological interventions to maximize outputs to meet the growing need for food for the increasing population [4]. Unfortunately, this increased food production also increased chemical pesticides and fertilizers, causing environmental and health issues.

Consumers worldwide are now more concerned with the environment [5]. They are sensitive to information about products, processing, and brands that might impact the environment [6]. Environmental issues are perceived as having a more direct impact on consumers' well-being. Consumers who know environmental degradation activities are willing to buy organic foods [7].

Heightened awareness of the environment and the consumer's desire to buy organic foods leads to increased corporate investment toward organic food production and marketing. They are thus initiating significant innovations in the organic food industry [8]. As a result, the organic food market is increasing [9]. In addition, effective campaigns create awareness about the environment. Because of these effective campaigns, consumers are now ready to spend more on green products [10].

Furthermore, people's living standards have significantly improved in the past few decades. With these improvements, the demand for better lifestyles and food has also increased. The steady growth in purchases of organic foods is an emerging trend. Consumers want to learn what organic foods offer before purchasing decisions [11].

**Global organic food market.**   According to a recent report, the organic food market is expected to grow with a Compound Annual Growth Rate (CAGR) of 16% during 2015–2020. This growth might be due to consumers' health concerns as they become aware of organic foods' perceived health benefits. Further, rising income levels, changes in living standards, and government initiatives encourage the broader adoption of organic products [12].

## Organic food market in the USA

In 2018, organic market sales were US$47.86 billion, and the market grew by 6.3% from 2017 to 2018 [13]. In 2017, the organic food market in the United States hit a record of US$45.2 billion in sales; this market consists of both the organic food market and the organic non-food market (see Fig 1). It is predicted that the organic food market will grow at a consistent pace as it matures. The demand for organic foods is flourishing as consumers seek nutritious and clean eating, which they perceive as suitable for their health and the environment.

Understanding consumer buying behavior toward organic foods is essential to pursue better marketing and management of the market. This can help us learn about the consumer decision-making process on organic foods and understand how consumers' attitudes and beliefs impact their consumption patterns. In addition, studying consumers' willingness to pay a premium price and their response to organic food advertisements [14] is necessary for companies to succeed in this growing market.

This study focuses on exploring the factors influencing consumers' buying behavior of organic foods. Although many factors can affect consumer buying behavior, we chose health consciousness, knowledge, subjective norms, price, and availability for this study based on Singh & Verma's [1] study. Understanding these factors is vital for developing marketing strategies for successfully marketing of these products.

## Theory and research hypotheses

Earlier research in the area of consumer buying behavior of organic foods discussed reasons why people buy. Even though there are some differences, the main reasons are product quality, concerns related to environmental degradation, and health-related issues [15]. Subsequent studies on consumer buying behavior of organic foods confirmed this [16]. Consumers tend to

Organic food and non-food sales in the United States from 2008 to 2018 (in billion U.S. dollars)
Organic food and non-food sales in the U.S. 2008-2018

**Fig 1. Organic food and non-food sales in the United States from 2008 to 2018 (in billions of US dollars).** Source: Statista.com.

perceive organic foods as being healthier than conventional alternatives. This perception of organic foods is one of the most commonly cited reasons for purchasing them. In two studies [17, 18], it became evident that consumers tend to have a positive attitude toward organic foods. However, they may not be purchasing organic foods due to environmental concerns. Instead, purchasing decisions are driven by the perceived health benefits the foods offer, the desire to fit in with a social group, try a new trend, or differentiate themselves from others [19].

## Health consciousness (HEC)

Consumer attitudes are significantly influenced by their health consciousness [20]. Consumers mainly purchase organic foods due to health benefits [21]. Several studies show that health factors significantly influence consumers' willingness to buy organic foods [22–26]. One of the significant reasons that influence consumers could be the deterioration of their health [22]; thus, consumers see consumers' purchases as an investment for good health. Bourn and Prescott [27] found that organic foods have a competitive advantage over conventional foods due to organic foods' nutritive attributes.

However, in a study conducted by Fotopoulos and Krystallis [28], taste is also another reason consumers buy organic foods. Even though many studies said that the perceived health benefits are the primary motivator, work by Tarkiainen and Sundqvist [29] and Michaelidou and Hassan [25] did not find it to be a compelling driver. In the earlier studies, the health benefit is the least significant influencer on organic foods. We examined our respondents' thoughts on this topic with these different findings on the importance of health benefits. Based on the above, we formulated Hypothesis 1:

**H1:** Health consciousness has a positive impact on buying behavior toward organic foods.

## Consumer knowledge (CK)

The Theory of Reasoned Actions (TRA) supports our understanding of consumer behavior development by exploring the motivational influences on how consumers behave [30]. TRA offers a basis for predicting consumer attitudes and behavior [31]. Liu [20] further confirms that TRA is the best theory to predict consumer behavior about organic foods. Consumers want to be aware of what they are buying and satisfy their needs and wants. Therefore, knowledge is essential in impacting consumer behavior on foods.

Sapp [32] argued that knowledge involves a cognitive learning process. Consumer purchase intentions differ based on the consumers' levels of expertise [33]. Consumers' purchase of organic products cannot be separated from their knowledge and understanding of organic foods [34, 35]. Recent research on consumer awareness and knowledge about organic foods found that consumer awareness worldwide is low relative to Europe's awareness level. This elevated awareness about organic food is due to its market, which is well developed compared to the rest [3, 36–39].

Studies also found that consumers' knowledge about what is "organic" is inconsistent. For example, in one study, respondents assumed that organic foods are produced without pesticides, fertilizers, or growth regulators [40]. However, in a similar study done in the UK by Hutchings and Greenhalgh [41], respondents thought that "organic" farming is free from chemicals and is grown naturally. Further, respondents felt that organic foods are not intensively farmed.

In consumer purchase decisions of organic foods, awareness and knowledge about these products are essential. Smith and Paladino [42] conducted a study on factors affecting organic foods' purchasing behavior. They found that learning about social and environmental issues will positively impact consumers' purchase behavior. However, from the above, it is evident that consumers' knowledge about organic foods is inconsistent. While they are likely to perceive that organic foods are pure, natural, and healthy, this perception might be based on their belief that organic foods are free from pesticides and chemical fertilizers. To evaluate the same, we proposed Hypothesis 2 as:

**H2:** Consumer buying behavior is positively associated with consumer knowledge of organic foods.

## Perceived or subjective norms (PSN)

Ajzen [43] defines perceived or subjective norms (PSN) as "a perceived social pressure to perform or not to perform a behavior." Finlay et al. [44] said subjective norms are individuals' perceptions or opinions about what others believe the individual should do. Subjective norms had an impact on consumer purchase behavior in the research conducted by Shimp and Kavas [45], Sheppard et al. [46], and Bagozzi et al. [47]. Chang [48] tested the correlation between attitudes toward consumer behavior and subjective norms. This study also examined the link between norms and attitudes and found that subjective norms lead to behavior attitudes in a meaningful manner. From the above, we formulated Hypothesis 3 as:

**H3:** Perceived or subjective norms will positively influence consumer buying of organic foods.

## Perception of the price (PP)

Organic foods are priced higher than conventional foods. Aertsens et al. [49] and Hughner et al. [16] confirmed that price is a significant barrier to organic food choice. Padel and Midmore [50] and O'Doherty et al. [51] indicate that high prices are likely to impede future demand development; thus, price is crucial in organic food marketing. The research confirmed that consumers switch products due to high prices [52], and Gan et al. [53] found that higher costs hurt the chances of buying organic foods. However, Radman [54] concluded that some consumers have a positive attitude toward organic foods and are willing to pay a higher price. Meanwhile, Smith et al. [55] found that price does not significantly impact organic food purchases. Since there are contradictory findings on the relationship between price and organic foods, we decided to explore whether consumer perceptions of cost have any link to their buying behavior of organic foods, as stated in Hypothesis 4:

**H4:** Perceived price of organic foods is negatively associated with consumer buying.

## Availability of organic foods (AV)

Availability is one factor that encourages the purchase of organic foods [56]. Makatouni [24] reiterated that organic foods' availability could be a barrier to consuming the same. In a study by Tarkiainen and Sundqvist [29], the authors showed that the easy availability of organic foods positively affected their purchase behavior. In a survey conducted by Young et al. [57], consumers prefer readily available products. Therefore, they do not want to spend time searching for organic products.

However, recently, retailers across the country have noticed the growing popularity of organic foods and have been adding organic foods to their shelves. Increased organic foods marketing by large retail outlets and specialty stores has made organic foods accessible to more consumers [58]. This discussion poses a question. Does availability have a positive impact on purchase behavior? We decided to test this using Hypothesis 5:

**H5:** Availability of organic foods increases consumer buying behavior.

## Purchase intention and actual buying behavior (PI and AB)

Planned behavior theory suggests that a reaction is a function of intentions and perceived behavioral control. Sheppard et al. [31] showed evidence that a relationship exists between choices and actions in different buying behavior types. Ajzen [43] stated that intentions or willingness could significantly predict actual buying behavior. Studies by Tanner and Kast [59] and Vermeir and Verbeke [60] found discrepancies between consumers who expressed favorable attitudes and actual purchase behavior. Hughner [16] found that, even though consumers have a positive attitude toward purchasing organic foods, very few people bought them. Based on the above, researchers believe that there is a relationship between attitudes and actions. This is in line with the study of Wheale and Hinton [61]. Attitudes toward organically grown food products might positively and significantly affect purchase behavior [62]. From this, it is assumed that the purchase of organic food results from an intent to purchase.

The attitude-behavior gap is a gap in consumers' favorable attitude and actual purchase behavior of organic foods. This gap suggests that a positive attitude toward organic products might not always lead to a purchase. Many factors could influence this gap. Price, availability, and social influence, among many others, can create a discrepancy among consumer attitudes, purchase intentions, and actual buying behavior. We test the effects of influencing factors (HEC, CK, PSN, PP, AV) on purchase intent (PI) and actual buying behavior (AB).

**H6:** Consumer attitudes toward organic foods mediate the association between influencing factors and purchase intention.

**H7:** Consumer attitudes and purchase intentions mediate the association between influencing factors and actual buying behavior.

## Sociodemographic factors

Behavior is not influenced by attitudes alone; many factors influence behavior. For example, Voon et al. [62] found that sociodemographic factors influence buying behavior. One significant factor is gender. For instance, Lockie et al. [63] confirm that women are more likely to have positive attitudes than men toward organic foods. Similarly, adolescent girls are more favorable than boys toward organic products [64].

Research has found that age also influences the purchase of organic foods. For example, Misra et al. [65] show that older individuals may be willing to buy organic foods due to health-related reasons. However, Cranfield and Magnusson [66] found that younger consumers are more likely to pay over 6% higher premiums to ensure that food products are pesticide-free. In addition, Rimal et al. [67] found that older individuals are less likely to buy organic foods than younger individuals. In contrast, younger people and women consider organic foods more essential and include them in their purchases [68, 69].

In consumers' demographic characteristics, income is another factor considered crucial for influencing the purchase of organic food. In two studies conducted by Govidnasamy and Italia [68] and Loureiro et al. [70], organic products are more frequently purchased by higher-income households. Likewise, Voon et al. 's [62] research found that household income positively relates to organic food purchases. Further, women in the 30–45, with children and having a higher disposable income, include organic foods in their purchases [58].

Research by Cunningham [38] and O'Donovan and McCarthy [71] found a positive relationship between organic foods and education consumption. This is also true of Dettmann and Dimitri's [58] work. According to their study, individuals with a higher education level are more likely to purchase organic foods than those with a lower education level. This was also discovered by Aryal et al. [72]. They showed that education is another factor that might influence the purchase of organic products.

Contrary to the above-referred research, some studies found a negative correlation [73, 74]. These negative correlations are also confirmed by the analysis of Arbindra et al. [75]. They explain that organic food purchase patterns and levels of education are statistically significant.

Since there are different findings in the literature, we test the influence of demographic factors on buying, and the following hypotheses are formulated:

**H8a:** *The age* of the consumer and buying behavior toward organic foods are significantly different.

**H8b:** *Gender* and buying behavior toward organic foods are significantly different.

**H8c:** *Income* and buying behavior toward organic foods are significantly different.

**H8d:** *Education* and buying behavior toward organic foods are significantly different.

## Research method

Primary data were collected using a questionnaire developed from prior studies [1, 76–80]. The questionnaire has two sections. The first section contains questions about organic product purchase behavior, with responses measured on a 5-point Likert scale. The second section includes questions on respondents' demographic information (see S1 Appendix).

The questionnaire was pilot tested on 50 respondents to ensure question and response clarity. Changes were made where necessary based on the feedback of the pilot study. Convenience and snowball sampling methods were used. Online surveys were conducted by sending out the surveys to individuals known to both the researcher and the students taking a Market Research course during Spring 2019. These individuals were asked to pass on the survey to their friends and family members. The snowball sampling method was used to generate as many responses as possible during May-August 2019. Respondents were asked to participate in the study via email. The email sent to potential participants indicated that they voluntarily agreed to participate in the survey by clicking on the survey link. The email also mentioned that, at any time, they could stop participating by merely closing the browser, and their responses will not be saved. A total of 770 responses were received. After going through the questionnaires for

completeness, a total of 502 surveys were used for further analysis. The study is approved by the Institutional Review Board of the University of Wisconsin-Stout as this involves a survey from the consumers based on their consents. Further, the data were analyzed anonymously.

## Results and discussion

The respondents' demographic profile is reported in Table 1. The table indicates that 58% of the respondents are men, while the remaining 42% are women. The plurality (37%) of the respondents is 31–40 years old. Likewise, most (35%) are graduate students, followed by undergraduate students (28%) and postgraduates/Ph.D. (21%). The analysis also shows that respondents' plurality has an annual income of over $100,000. The highest proportion of respondents (38%) has a family size of 1–2 members living in their households. This family size is closely followed by 3–4 people in the household (37%).

### Reasons for purchase of organic foods

Respondents were asked if they have ever bought organic food products, and 55.6% said yes. Then, these respondents were asked further questions about their purchases. When asked about the purchase frequency, 51.8% said they purchase organic food products weekly, 26% purchase at least once a month, and the remaining 21.6% purchase less frequently than once a month.

Respondents mentioned health consciousness as the primary reason for purchasing organic food. Further, non-use of pesticides, lower pesticide residues, environmentally friendly production, and perceived freshness are other reasons respondents choose to buy organic foods (see Fig 2). Health consciousness played an essential role in 48% of respondents, followed by pesticide-free (19%) and environmentally friendly (15%) considerations.

To identify the factors influencing attitudes toward organic foods, Principal Components Analysis (PCA) using varimax rotation is conducted. Before applying the factor analysis, the Kaiser-Mayer-Olin (KMO) test and Bartlett's test of sphericity are used to test data suitability.

**Table 1. Sociodemographic characteristics of respondents.**

| Characteristic | N (%) | Characteristic | N (%) |
|---|---|---|---|
| **Gender** | | **Family Annual Income** | |
| Male | 291 (58) | Less than $40,000 | 45 (9) |
| Female | 211 (42) | $40,001 to $60,000 | 80 (16) |
| | | $60,001 to $80,000 | 116 (23) |
| **Age** | | $80,001 to $100,000 | 126 (25) |
| 18–30 years | 105 (21) | above $100,000 | 135 (27) |
| 31–40 years | 186 (37) | | |
| 41–50 years | 116 (23) | **Family Size** | |
| 51–60 years | 60 (12) | 1–2 | 191 (38) |
| Above 60 years | 35 (7) | 3–4 | 186 (37) |
| | | 5 or more | 125 (25) |
| **Education** | | | |
| High school | 80 (16) | **Occupation** | |
| Undergraduate | 141 (28) | Student | 55 (11) |
| Graduate | 176 (35) | Work full-time | 186 (37) |
| Postgraduate/Ph.D. | 105 (21) | Self-employed | 171 (34) |
| | | Retired | 90 (18) |

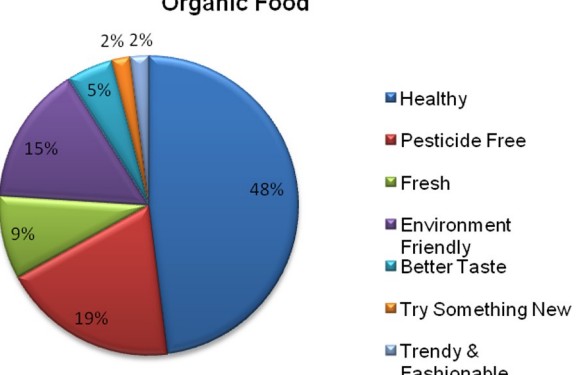

**Fig 2. Reasons for purchasing organic foods.**

The result shows the KMO measure of sampling adequacy as 0.82. Thus, the value exceeds the cut-off value of 0.60. Bartlett's test of sphericity ($\chi^2$ = 2,082, df = 132, $p <$ .001) is also significant. This indicates that the inter-item correlations are significant for PCA. KMO and Bartlett's test results support the data [81]. The results are shown in Table 2 to ensure scale reliability. Each factor has a Cronbach's alpha ($\alpha$) value higher than the threshold value of 0.70 [82].

Multiple linear regression analysis is performed to test hypotheses H1–H5. The analysis ascertains the impact of health consciousness, consumers' knowledge, perceived or subjective norms, availability, and perception of the price on consumer attitude (AT). As shown in Table 3, HEC, CK, PSN, PP, and AV account for 33% of the explained variances (F (5, 177) = 32.51, $p <$ .001, $R^2$ = 0.33).

According to the results, the H1($\beta$ = 0.37, $p$ = .016); H2 ($\beta$ = 0.47, $p <$ .001); H3 ($\beta$ = 0.34, $p$ = .015); and H4 ($\beta$ = 0.36, $p$ = .001) are supported, as the $\beta$ values are positive and significant. However, the values for H5 ($\beta$ = 0.29, $p$ = .117) are statistically non-significant. This shows that H5 is not supported. The findings confirmed that health consciousness, consumer knowledge, perceptive or subjective norm, and perception of the price affect respondents' attitudes toward organic foods. However, it is also found that availability has no impact on consumers' attitudes, at least in our sample.

The hierarchical regression method was applied to test the association between purchase intention and influencing factors (HEC, CK, PSN, PP, and AV) via the mediation of AT. The mediation was ascertained using Baron and Kenny's [83] approach. Certain criteria must be met to declare the presence of mediation in the equation. The first necessary criterion is that the independent variable (IV) must affect the dependent variable (DV). The second criterion is that the IV must significantly influence the mediating variables. The third suggests the mediating variables must affect the DV. When all of the above conditions are met, a full mediation is confirmed if the IV no longer affects the DV after the mediator has been controlled for. Partial mediation occurs when the effect of the IV on the DV is reduced after the mediators are controlled for. The results indicate that all $\beta$ values (for the effect on AT) are positive and significant: HEC ($\beta$ = 0.17, $p <$ .001), CK ($\beta$ = 0.29, $p <$ .030), PSN ($\beta$ = 0.33, $p <$ .020), PP ($\beta$ = 0.39, $p <$ .010), and AV ($\beta$ = 0.24, $p <$ .050; see Table 4). The presence of mediation is also confirmed, as Baron and Kenny's criteria are met. Thus, H6, which predicts that the attitude mediates the relationship between the influencing factors and PI, is supported.

According to the results reported in Table 5, H7—which states that influencing factors have a positive effect on actual buying behavior via the mediating effect of attitude and purchase intention—is supported: AT ($\beta$ = 0.24, $p <$ .040) and PI ($\beta$ = 0.26, $p <$ .020). This confirms

**Table 2. Constructs, observable items, and factor loadings.**

| Construct | Indicator | Factor Loadings (λ) | Cronbach's α | Variance (%) |
|---|---|---|---|---|
| Health Consciousness | | | 0.78 | 37.12 |
| | HEC1 | 0.81 | | |
| | HEC2 | 0.78 | | |
| | HEC3 | 0.82 | | |
| Consumers' Knowledge | | | 0.79 | 7.63 |
| | CK1 | 0.73 | | |
| | CK2 | 0.84 | | |
| | CK3 | 0.92 | | |
| Perceived or Subjective Norms | | | 0.87 | 5.60 |
| | PSN1 | 0.81 | | |
| | PSN2 | 0.86 | | |
| | PSN3 | 0.76 | | |
| Perception Price | | | 0.84 | 5.38 |
| | PP1 | 0.86 | | |
| | PP2 | 0.83 | | |
| Availability of Organic Foods | | | 0.70 | 3.76 |
| | AV1 | 0.81 | | |
| | AV2 | 0.76 | | |
| Attitude | | | 0.84 | 3.46 |
| | AT1 | 0.81 | | |
| | AT2 | 0.77 | | |
| | AT3 | 0.87 | | |
| Purchase Intention | | | 0.79 | 2.13 |
| | PI1 | 0.91 | | |
| | PI2 | 0.77 | | |
| | PI3 | 0.83 | | |
| Actual Buying Behavior | | | 0.82 | 1.69 |
| | AB1 | 0.91 | | |
| | AB2 | 0.74 | | |
| | AB3 | 0.87 | | |

that AT and PI have a positive and significant effect on consumers' actual buying behavior. Furthermore, AT and PI mediate the association between influencing factors and AB since the values of the corresponding regression coefficients of HEC, CK, PSN, PP, and AV are reduced when the effects of AT and PI are controlled for. These results support H7.

**Table 3. Results of multiple regression analysis on consumer attitudes.**

| Predictor | Min | Max | Mean | SD | B | Regression Analysis | | | Collinearity | |
|---|---|---|---|---|---|---|---|---|---|---|
| | | | | | | SE | t | Sig | TOL | VIF |
| HEC | 1 | 5 | 4.23 | 1.03 | 0.37 | 0.06 | 6.14 | .016 | 0.84 | 1.20 |
| CK | 1 | 5 | 3.84 | 1.12 | 0.47 | 0.13 | 5.27 | .000 | 0.67 | 1.50 |
| PSN | 1 | 5 | 4.44 | 1.66 | 0.34 | 0.18 | 3.56 | .015 | 0.51 | 1.96 |
| PP | 1 | 5 | 4.06 | 1.45 | 0.36 | 0.15 | 5.25 | .001 | 0.43 | 2.30 |
| AV | 1 | 5 | 3.74 | 1.12 | 0.29 | 0.08 | 2.41 | .117 | 0.49 | 2.04 |

Notes: $R^2 = 0.33$, $F (5, 177) = 32.51$

Table 4. Results of purchase intention predictions using hierarchical regression analysis.

| Predictor | Step 1 | | | Step 2 | | | Collinearity | |
|---|---|---|---|---|---|---|---|---|
| | B | t | Sig | B | T | Sig | TOL | VIF |
| HEC | 0.26 | 6.14 | .030 | 0.17 | 5.10 | .000 | 0.72 | 1.40 |
| CF | 0.41 | 5.22 | .000 | 0.29 | 3.23 | .030 | 0.81 | 1.20 |
| PSN | 0.38 | 3.18 | .010 | 0.33 | 2.79 | .020 | 0.41 | 2.40 |
| PP | 0.41 | 6.99 | .010 | 0.39 | 6.18 | .010 | 0.38 | 2.60 |
| AV | 0.31 | 7.67 | .040 | 0.24 | 3.55 | .050 | 0.48 | 2.10 |
| AT | | | | 0.39 | 4.67 | .020 | 0.53 | 1.90 |
| R2 | 0.43 | | | 0.46 | | | | |

## Demographic differences in the actual buying behavior

An independent t-test is conducted to see if the actual purchase behavior changes are due to gender. Levene's test (Table 6) indicates that the p-value for gender is more significant than .05. The result confirms the homogeneous variance. Thus, the t-test is suitable for equal variance. Furthermore, the t-value of 0.08 (two-tailed) is higher than the significance level, suggesting a non-significant difference, implying that the mean values (-0.19 and -0.16) are not significant, supporting H8a.

Table 7A below shows the results of the one-way ANOVA test. The findings suggest that respondents' age (F = 7.01; $p$ = .023) has a statistically significant effect on the purchase intention; thus, H8b is supported. However, further analysis of the respondents' age groups is conducted using the least significant difference (LSD) test. The results of the LSD test, as depicted in Table 7B, indicate that the age group of 41–50 years has a statistically higher score than other age groups.

Hypothesis H8c is supported, as the ANONA test reveals that annual income (F = 8.22; $p$ = .011) significantly affects purchase intention (see Table 8A). Further, the LSD Test for income (Table 8B) implies that the income level of more than US$80,000 has a higher score on the actual purchase as compared to those with incomes lower than US$80,000.

According to Table 9A, the level of education (F = 7.05; p = .001) affects consumer purchase behavior toward organic foods. The LSD test (Table 9B) further clarifies that consumers hold postgraduate/Ph.D. Degrees have a higher score on the AB of organic food products than

Table 5. Results of purchase intention predictions using hierarchical regression analysis.

| Predictor | Step 1 | | Step 2 | | Step 3 | | Collinearity | |
|---|---|---|---|---|---|---|---|---|
| | β | t | B | T | B | T | TOL | VIF |
| HEC | 0.27 | 4.28* | 0.63 | 5.54* | 0.22 | 5.44* | 0.45 | 2.22 |
| CK | 0.35 | 3.25* | 0.44 | 4.77* | 0.43 | 4.22* | 0.35 | 2.86 |
| PSN | 0.39 | 4.18* | 0.34 | 2.33* | 0.33 | 4.19* | 0.41 | 2.44 |
| PP | 0.48 | 5.77* | 0.55 | 5.21* | 0.45 | 6.75* | 0.39 | 2.56 |
| AV | 0.43 | 0.33* | 0.27 | 4.81* | 0.33 | 7.67* | 0.43 | 2.33 |
| AT | | | 0.46 | 5.23* | 0.24 | 5.39* | 0.49 | 2.04 |
| PI | | | | | 0.26 | 7.67* | 0.34 | 2.94 |
| ΔR2 | 0.55 | | 0.07 | | 0.08 | | | |
| ΔF | 117.77 | | 18.55 | | 22.77 | | | |

Notes: * $p$ < .05;

** $p$ < .001

**Table 6. Gender: Independent t-test.**

| | Levene's Test for Equality of Variances | | T-Test for Equality of Means | | | | | | 95% Confidence Interval of the Difference | |
|---|---|---|---|---|---|---|---|---|---|---|
| | F | Sig. | T | Df | Sig. (2-tailed) | Mean Difference | Std. Error Difference | | | |
| | | | | | | | | | Lower | Upper |
| Equal Variances | 2.61 | .153 | -1.21 | 500 | .081 | -.19 | .07 | | -.32 | -.01 |
| Equal Variances not assumed | | | -1.08 | 472.55 | .069 | -.16 | .08 | | -.32 | -.01 |

consumers with only a high school diploma or undergraduates. The test also shows that graduate degree-holders are more likely to purchase organic food than any other group.

## Conclusions

This study tested Singh and Verma's [1] model on US consumers. We initially investigated the factors influencing consumer attitudes. Then we studied how these influencing factors and attitudes together affect the actual buying behavior of consumers. There has always been a debate on consumers' intention to purchase compared to their actual purchase. Evidence of previous studies suggests that actual purchase behavior is not always the consequence of intent to purchase. Consumers sometimes intend to buy but often fail to do so. Therefore, this study also looked at the impact of demographic variables (such as gender, income, education, and age) on the consumers' actual buying. This study confirms that all five factors—namely, health consciousness, consumer knowledge, availability, perception of price, and subjective norms—influence consumer attitudes. In contrast, attitudes and purchases were found to have mediating roles between influencing factors and actual buying behavior toward organic foods.

Further, the t-tests and ANOVA test results explored a more in-depth understanding of the relationships between demographic factors and actual buying. LSD tests were conducted to understand which sub-group in a demographic variable is significantly different from its counterparts. The findings of this study suggest that gender does not affect the actual buying of organic foods. Meanwhile, income, age, and education do affect consumers' actual purchases. Furthermore, the LSD test shows that 41–50 years of age, consumers are more likely to buy organic foods than those in other groups. Not surprisingly, income is found to be another

**Table 7.** A. Age groups: ANOVA test. B. LSD test for respondent's age groups.

**A**

| Actual Buying Behavior | Sum of Squares | Df | Mean Square | F | p-values |
|---|---|---|---|---|---|
| Between Groups | 7.18 | 4 | 3.36 | 7.01 | .023 |
| Within Groups | 156.13 | 498 | .41 | | |
| Total | 163.31 | 502 | | | |

**B**

| Dependent Variable | | Respondent's age | Mean Difference | p-value |
|---|---|---|---|---|
| | (I) | (J) | (I–J) | |
| Actual Buying Behavior | | 18–30 years | 0.30 | .020 |
| | 41–50 years | 31–40 years | 0.21 | .000 |
| | | 51–60 years | 0.32 | .000 |
| | | above 60 years | 0.43 | .000 |

Notes:

1. p-values are rounded off to three decimal places.

2. Statistical significance is tested at p < 0.05.

**Table 8.** A. Annual income: ANOVA test. B. Annual income: LSD test.

**A**

| Actual Buying Behavior | Sum of Squares | df | Mean Square | F | Sig |
|---|---|---|---|---|---|
| Between Groups | 15.10 | 5 | 4.22 | 8.22 | .011 |
| Within Groups | 144.86 | 497 | 0.35 | | |
| Total | 159.96 | 502 | | | |

**B**

| Dependent Variable | Respondent's annual income ($) | | Mean Difference | Sig. |
|---|---|---|---|---|
| | (I) | (J) | (I–J) | |
| Actual Buying Behavior | 40,001 to 60,000 | Less than 40,000 | 0.47 | .000 |
| | | 40,001 to 60,000 | 0.43 | .011 |
| | | 80,001 to 100,000 | 0.37 | .023 |
| | 80,001 to 100,000 | Less than 40,000 | 0.61 | .042 |
| | | 40,001 to 60,000 | 0.44 | .000 |
| | | 60,001 to 80,000 | 0.41 | .000 |
| | | above 100,000 | 0.42 | .046 |
| | above 100,000 | Less than 40,000 | 0.60 | .018 |
| | | 40,001 to 60,000 | 0.53 | .000 |
| | | 60,001 to 80,000 | 0.47 | .000 |
| | | 80,001 to 100,000 | 0.41 | .030 |

Notes:

1. p-values are rounded off to three decimal places.

2. Statistical significance is tested at $p < 0.05$.

critical determinant of actual buying decisions. This may indicate that income is directly proportional to organic food buying (i.e., the higher the income level, the more likely the consumer is to buy organic foods). The findings also indicate the same trend with education. Higher levels of education correspond to a higher likelihood of purchasing organic foods. This

**Table 9.** A. Education levels: ANOVA Test. B. Education levels: LSD Test.

**A**

| Actual Buying Behavior | Sum of Squares | Df | Mean Square | F | p-value |
|---|---|---|---|---|---|
| Between Groups | 11.81 | 4 | 3.78 | 7.05 | .001 |
| Within Groups | 137.31 | 498 | 0.39 | | |
| Total | 142.12 | 502 | | | |

**B**

| Dependent Variable | Respondent's education level | | Mean Difference | p-value |
|---|---|---|---|---|
| | (I) | (J) | (I–J) | |
| Actual Buying Behavior | Graduate | High School | 0.40 | .000 |
| | | Undergraduate | 0.34 | .018 |
| | Postgraduate/Ph. D. | High School | 0.53 | .000 |
| | | Undergraduate | 0.51 | .210 |
| | | Graduate | 0.37 | .000 |

Notes:

1. p-values are rounded off to three decimal places.

2. Statistical significance is tested at $p < 0.05$.

could be because education might increase the consumer's knowledge, and informed consumers could be health-conscious and aware of organic foods' benefits. Many studies have stated different reasons for buying organic foods in developed and developing countries. However, if we compare and contrast our research findings with recent work in developed countries, similar results have been obtained. Health consciousness, food safety, environmentally friendly procedures, consumer's knowledge on organic foods, perceived or subjective norms, availability of organic foods, and demographic factors, like gender, education, and income are the most substantial reasons for buying organic food, irrespective of the country (developed or developing; [1, 3, 25].

## Implications

The findings of this research may guide companies dealing with organic foods. The study suggests the companies can craft marketing strategies to increase consumers' awareness of the benefits of organic food consumption. Providing additional information about the benefits of organic food products may help convince consumers to make the purchase. This study will be helpful to retailers to segment their consumers based on their demographics. The study will also help retailers understand the factors that are likely to influence consumers' organic food purchases and design strategies to increase their sales. Since availability (access) is one factor in buying decisions, retailers should reach out to local shops/areas to enhance market coverage. As subjective norms are another significant factor, marketers should promote organic food consumption through family, celebrities, and society.

This study offers important implications but with some limitations. First, direct factors related to consumer purchase decisions were measured. The second limitation is the sampling. Since the data is collected using an online survey forwarded by students and researchers to others, it could constitute snowballing. Any data collected using snowballing should be cautiously used to generalize the outcomes. Further research in this area may consider advertisements, federal and state regulations, and consumption patterns of organic foods. Of course, in organic food consumption, more studies in different regions with a higher sample size would validate our findings.

Covid-19 pandemic crisis affecting all aspects of the population's daily life, in particular, dietary habits [85]. However, Covid-19 perceptions on adopting healthy food habits are not investigated in the present study. Any further research in this area should consider post-pandemic behavior. Recent studies suggest that parental attitudes affects dietary habits [84–86]. Therefore, future research should also consider how parental attitudes influence the purchase of organic foods.

## Supporting information

**S1 Dataset.**
(XLSX)

**S1 Appendix. Survey questionnaire.**
(DOCX)

## Author Contributions

**Conceptualization:** Raghava R. Gundala, Anupam Singh.

**Data curation:** Anupam Singh.

**Formal analysis:** Anupam Singh.

**Investigation:** Raghava R. Gundala.

**Methodology:** Anupam Singh.

**Project administration:** Raghava R. Gundala.

**Software:** Anupam Singh.

**Writing – original draft:** Raghava R. Gundala, Anupam Singh.

**Writing – review & editing:** Raghava R. Gundala, Anupam Singh.

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
