## [Decision Letter · Decision Letter 0]

18 Jun 2021

PONE-D-21-00520

What Motivates Consumer to Buy Organic Foods? Results from an Empirical Study in Midwestern United States

PLOS ONE

Dear Dr. Singh,

Thank you for submitting your manuscript to PLOS ONE. After careful consideration, we feel that it has merit but does not fully meet PLOS ONE’s publication criteria as it currently stands. Therefore, we invite you to submit a revised version of the manuscript that addresses the points raised during the review process.

We look forward to receiving your revised manuscript.

Kind regards,

Ali B. Mahmoud, Ph.D.

Academic Editor

PLOS ONE

**Additional Editor Comments:**

Please revise the statistical methods employed in this study. For instance, the probability value is never equal to zero. More accurately, it is always above zero.COVID perceptions effects on adopting healthy food habits are not investigated. Thus, this must be addressed as a research limitation/implication by citing [1, 2]. Also, parental attitudes need to be highlighted for future research [1, 3]. 

**References**

 1. Mahmoud AB, Hack-Polay D, Fuxman L, Naquiallah D, Grigoriou N: **Trick or treat? – when children with childhood food allergies lead parents into unhealthy food choices**. *BMC Public Health *2020, **20**(1):1453.

2. Mahmoud AB, Hack-Polay D, Fuxman L, Nicoletti M: **The Janus-faced effects of COVID-19 perceptions on family healthy eating behaviour: Parent’s negative experience as a mediator and gender as a moderator**. *Scandinavian Journal of Psychology *2021.

3. Mahmoud AB, Grigoriou N: **Modelling parents’ unhealthy food choices for their children: the moderating role of child food allergy and implications for health policy**. *Journal of Family Studies *2019:1-19.

Journal Requirements:

4. We note that you have referenced (Yi, L. K. (2009) which has currently not yet been accepted for publication. Please remove this from your References and amend this to state in the body of your manuscript: Yi, L. K. (2009) Unpublished honours degree project as detailed online in our guide for authors

http://journals.plos.org/plosone/s/submission-guidelines#loc-reference-style. 

**Reviewers' comments:**

Reviewer's Responses to Questions

**Comments to the Author**

1. Is the manuscript technically sound, and do the data support the conclusions?

Reviewer #1: Yes

Reviewer #2: Yes

2. Has the statistical analysis been performed appropriately and rigorously? 

Reviewer #1: Yes

Reviewer #2: Yes

3. Have the authors made all data underlying the findings in their manuscript fully available?

Reviewer #1: Yes

Reviewer #2: Yes

4. Is the manuscript presented in an intelligible fashion and written in standard English?

Reviewer #1: Yes

Reviewer #2: Yes

5. Review Comments to the Author

Reviewer #1: Dear Author(s)

Congratulations on a rigorous study on an important and relevant topic. Here are some of my comments -

1) I am uncomfortable with the title where the focus seems to be on consumers from Mid-western US. I understand that the authors may be located in the area and it is acceptable to focus in that region. However, given that your title is specifically about Midwest consumers, there needs to be a detailed section in the Literature review on why you chose consumers from that area. Any stats on mid-west consumers' spending on organic food? Why are they important compared to the rest of US consumers. If you cannot justify this, then I am uncomfortable with the entire study and you should label the data as convenience sample.

2) The limitations of this study should include that the data was collected by means of a snowball sampling technique.

3) With respect to the survey questions, did you disqualify respondents if they did not "purchase organic products" (Q2)?

4) For Q2 above, you should have indicated a time frame on when they purchased organic products in the past? Why did you generalize as "organic products" and not "organic food"?

5) The first line under Conclusion indicates that the study tested the research model on US consumers. You cannot generalize your conclusions to all of US consumers if you are just focusing on Mid-west consumers

6) Provide justification for why gender differences were not significant in your study.

Reviewer #2: The manuscript is well written and well read on the consumer perception of organic food. The analysis taken was fresh, suitable and justified. The findings obtained contribute to the numerous study on consumer perception with the uniqueness of the assigned community of Midwestern United States addressed. Research ethics had also been addressed.

6. PLOS authors have the option to publish the peer review history of their article (what does this mean?). If published, this will include your full peer review and any attached files.

Reviewer #1: No

Reviewer #2: **Yes: **Farah Ayuni Shafie

---

## [Author Response · Author response to Decision Letter 0]

29 Jun 2021

We thank the editor and the reviewers for their very thoughtful comments on our paper. All the suggestions and queries of reviewers have been considered carefully. We have responded to each comment and made appropriate changes to the manuscript. Reviewers and editor’s comments are in bold, authors’ responses are in plain text. 

Reviewer#1

1) I am uncomfortable with the title where the focus seems to be on consumers from Mid-western US. I understand that the authors may be located in the area and it is acceptable to focus in that region. However, given that your title is specifically about Midwest consumers, there needs to be a detailed section in the Literature review on why you chose consumers from that area. Any stats on mid-west consumers' spending on organic food? Why are they important compared to the rest of US consumers. If you cannot justify this, then I am uncomfortable with the entire study and you should label the data as convenience sample.

Authors: We agree with the reviewer's comment on the title. We changed to title and removed the word 'mid-western. Since there is no difference with the rest of the US consumers, it isn't easy to justify this. The data details are changed to a convenience sample. 

2) The limitations of this study should include that the data was collected by means of a snowball sampling technique.

Authors: It is added to the limitations.

3) With respect to the survey questions, did you disqualify respondents if they did not "purchase organic products" (Q2)?

Authors: No. Respondents who did not purchase are also considered for the survey. We are interested in knowing what factors might influence them to purchase in the future.

4) For Q2 above, you should have indicated a time frame on when they purchased organic products in the past? Why did you generalize as "organic products" and not "organic food"?

Authors: The reason for not adding a time frame is to avoid 'recall error.' It might be difficult for the respondent to remember the time frame of purchase. 

Authors: Regarding the “organic products”, we are extremely sorry that old version of questionnaire was appended. Intially, we were interested in studying the consumer buying behaviour toward organic products. But, during the pilot survey we learned that “organic products” would be a too broad term which includes clothing and personal care items, and consumers may view or behave differently for different items such as food, clothing and personal care. Therefore, in the final survey, we used the term ‘organic food products’. We thank the reviewer for highlighting it. The final version of questionnaire is appended in the revised manuscript. 

5) The first line under Conclusion indicates that the study tested the research model on US consumers. You cannot generalize your conclusions to all of US consumers if you are just focusing on Mid-west consumers.

Authors: Noted, and the changes are made in the document by removing the word 'mid-western consumers.'

Reviewer#2

Authors: Thank you for the feedback.

Response to editor

1) Please revise the statistical methods employed in this study. For instance, the probability value is never equal to zero. More accurately, it is always above zero.

Authors: We agree with the editor that p-value (probability) is always above zero. Using SPSS software, in our tables, p-values are rounded off to three decimal places. Using built-in rounding rules, it was rounded down and reported as .000. The value reported as .000 is not absolute zero or exctally zero. The probability (p-value) is very small, however, there is a still some chances of occurance of the event. We have provided notes to the table in the revised manuscript. Thanks for highlighting it. 

2) COVID perceptions effects on adopting healthy food habits are not investigated. Thus, this must be addressed as a research limitation/implication by citing [1, 2]. Also, parental attitudes need to be highlighted for future research [1, 3].

Authors: Noted. We have incorporated the suggestions in the revised manuscript.

---

## [Decision Letter · Decision Letter 1]

31 Aug 2021

What Motivates Consumers to Buy Organic Foods? Results of an Empirical Study in the United States

PONE-D-21-00520R1

Dear Dr. Singh,

We’re pleased to inform you that your manuscript has been judged scientifically suitable for publication and will be formally accepted for publication once it meets all outstanding technical requirements.

Kind regards,

Ali B. Mahmoud, Ph.D.

Academic Editor

PLOS ONE

Additional Editor Comments (optional):

Reviewers' comments:

Reviewer's Responses to Questions

**Comments to the Author**

1. If the authors have adequately addressed your comments raised in a previous round of review and you feel that this manuscript is now acceptable for publication, you may indicate that here to bypass the “Comments to the Author” section, enter your conflict of interest statement in the “Confidential to Editor” section, and submit your "Accept" recommendation.

Reviewer #1: All comments have been addressed

Reviewer #2: (No Response)

2. Is the manuscript technically sound, and do the data support the conclusions?

Reviewer #1: Yes

Reviewer #2: Yes

3. Has the statistical analysis been performed appropriately and rigorously? 

Reviewer #1: Yes

Reviewer #2: Yes

4. Have the authors made all data underlying the findings in their manuscript fully available?

Reviewer #1: Yes

Reviewer #2: Yes

5. Is the manuscript presented in an intelligible fashion and written in standard English?

Reviewer #1: Yes

Reviewer #2: Yes

6. Review Comments to the Author

Reviewer #1: (No Response)

Reviewer #2: (No Response)

7. PLOS authors have the option to publish the peer review history of their article (what does this mean?). If published, this will include your full peer review and any attached files.

Reviewer #1: **Yes: **Archana Kumar

Reviewer #2: **Yes: **Farah Ayuni Shafie

---

## [Editor Report · Acceptance letter]

3 Sep 2021

PONE-D-21-00520R1 

What Motivates Consumers to Buy Organic Foods? Results of an Empirical Study in the United States 

Dear Dr. Singh:

I'm pleased to inform you that your manuscript has been deemed suitable for publication in PLOS ONE. Congratulations! Your manuscript is now with our production department. 

Kind regards, 

on behalf of

Dr. Ali B. Mahmoud 

Academic Editor

PLOS ONE